# Dynamic chloride ion adsorption on single iridium atom boosts seawater oxidation catalysis

Xinxuan Duan[1,2,18], Qihao Sha[1,18], Pengsong Li[3], Tianshui Li[1], Guotao Yang[1], Wei Liu[1], Ende Yu[4], Daojin Zhou[1], Jinjie Fang[5], Wenxing Chen [6], Yizhen Chen [7], Lirong Zheng[8], Jiangwen Liao [8], Zeyu Wang [9], Yaping Li[1], Hongbin Yang[10], Guoxin Zhang[11], Zhongbin Zhuang [5,12], Sung-Fu Hung [13], Changfei Jing[14], Jun Luo [15], Lu Bai[16], Juncai Dong [8], Hai Xiao [9], Wen Liu [1], Yun Kuang[1,4] ✉, Bin Liu [10,17] ✉ & Xiaoming Sun [1] ✉

Seawater electrolysis offers a renewable, scalable, and economic means for green hydrogen production. However, anode corrosion by Cl⁻ pose great challenges for its commercialization. Herein, different from conventional catalysts designed to repel Cl⁻ adsorption, we develop an atomic Ir catalyst on cobalt iron layered double hydroxide (Ir/CoFe-LDH) to tailor Cl⁻ adsorption and modulate the electronic structure of the Ir active center, thereby establishing a unique Ir-OH/Cl coordination for alkaline seawater electrolysis. *Operando* characterizations and theoretical calculations unveil the pivotal role of this coordination state to lower OER activation energy by a factor of 1.93. The Ir/CoFe-LDH exhibits a remarkable oxygen evolution reaction activity (202 mV overpotential and TOF = 7.46 $O_2\ s^{-1}$) in 6 M NaOH+2.8 M NaCl, superior over Cl⁻-free 6 M NaOH electrolyte (236 mV overpotential and TOF = 1.05 $O_2\ s^{-1}$), with 100% catalytic selectivity and stability at high current densities (400-800 mA cm⁻²) for more than 1,000 h.

Grid-scale water electrolysis is promising for storing renewable electricity into molecular hydrogen bond[1-4]. Using earth-abundant seawater as feedstock instead of desalinated water provides a more sustainable strategy for renewable hydrogen production[5-7], which yields NaCl as a byproduct at the same time[8]. However, seawater has a salinity of ~3.5 wt%, in which most of the salt is NaCl (~0.5 M). Implementation of seawater electrolysis technology confronts many challenges, especially at the anode side where severe catalyst corrosion and competitive chloride oxidation reaction (ClOR) occur simultaneously[9-12], which significantly hinder its commercialization. Efficient and sustained seawater electrolysis demands a highly active and selective anode that is able to work in the presence of concentrated Cl⁻ [13,14]. Previous attempts of seawater electrolysis mostly focused on the prohibition of Cl⁻ adsorption on the anode catalyst so as to prevent ClOR. For instance, Koper et al.[15] electrodeposited MnOx on IrO₂ to improve the oxygen evolution reaction (OER) selectivity in acidic seawater electrolysis because of the weak chloride binding on MnOx surface. Ni, Co, Fe-based sulfides[16], phosphides[17], selenides[18], and boron-modified cobalt iron layered double hydroxides[19] showed anti-corrosion ability in alkaline seawater electrolysis due to the in situ formed Cl⁻ repelling anion or polyanion layer. Unfortunately, this also weakened the adsorption of O-intermediates of OER, which might decrease the OER activity. Additionally, at high overpotential (i.e., high current density), the Cl⁻ repelling strategy becomes less effective due to the enhanced driving force of Cl⁻ adsorption[20-22].

Herein, we design an atomic Ir catalyst on cobalt iron layered double hydroxide (Ir/CoFe-LDH) for electrochemical seawater oxidation. Different from conventional catalysts designed to completely repel Cl⁻ adsorption for seawater electrolysis, the atomic Ir sites on CoFe-LDH allow Cl⁻ adsorption to modulate the electronic structure of

Ir active center. As a result, the Ir/CoFe-LDH exhibits a remarkable OER performance in alkaline seawater electrolysis with an overpotential as low as 202 mV at the current density of 10 mA cm$^{-2}$, 34 mV lower than that in NaOH electrolyte. Moreover, the Ir/CoFe-LDH affords a remarkable activity at industrial-relevant current densities (0.4−0.8 A cm$^{-2}$) with close to 100% oxygen Faradaic efficiency for more than 1000 h. Meanwhile, it can also operate stably in real seawater, reaching 10 mA cm$^{-2}$ with an overpotential of 208 mV, and maintains stable operation for more than 2000 h at a current density of 1 A/cm$^2$. Both in situ experiments and theoretical analyses show that the dynamic chloride ion adsorption on single Ir atom during OER can effectively reduce the energy barrier to form *OOH (the rate-determining step of OER) and thus boost water oxidation catalysis, while at the same time maintaining a high energy barrier for the competitive ClOR.

## Results and discussion

### Synthesis and structural characterization of Ir/CoFe-LDH

Iridium-based catalyst has been widely used in catalyzing water electrolysis[23–26] and chloralkaline process[27–29], in which the Ir-chloride bond strength determines its selectivity[30–32]. LDH elemental combinations with lower electronegativity may exhibit enhanced electron coupling with metal single atoms[33]. Based on electronegativity values (Fe (1.83) < Co (1.88) < Ni (1.92)), it is anticipated that noble metal single atoms on CoFe LDHs may demonstrate superior OER performance. In this work, we develop a two-step synthesis to prepare single Ir atom catalysts with tunable single atomic Ir coordination structure on layered double hydroxide (LDH), as shown in Supplementary Fig. 1. In the first step, CoFe-LDH was synthesized through a co-precipitation method. Subsequently, dilute solutions of IrCl$_3$ and NaOH were added to the homogeneous LDH colloid to anchor atomic Ir onto CoFe-LDH as well as tune the chloride bonding on the single atomic Ir sites. SEM images reveal that both CoFe-LDH and Ir/CoFe-LDH exhibit uniform nanosheet structures, with no noticeable changes in surface morphology upon Ir loading (Supplementary Fig. 2). The Ir in the as-prepared Ir/CoFe-LDH catalyst was determined to be -0.5 wt% by inductively coupled plasma mass spectrometry (ICP-MS). Transmission electron microscopy (TEM) characterization shows clean surface of Ir/CoFe-LDH without formation of nanoparticles/nanoclusters (Supplementary Fig. 3). The high angle annular dark field scanning transmission electron microscope (HADDF-STEM) images clearly show bright spots, which can be assigned to single Ir atoms anchored on the surface of CoFe-LDH (Fig. 1a). Meanwhile, Ir atoms still show isolated bright spots at the same position in the HAADF-STEM images with a 30° tilt (Supplementary Fig. 4), confirming single atomic dispersion of Ir. There seems to be a cluster in the yellow circle in Supplementary Fig. 5a and in the red circle in Supplementary Fig. 5b, however, they both display single atoms at other viewing angles, indicating that the "agglomeration" of Ir single atoms may just be caused by viewing angles. Dr. probe STEM simulation software[34] was used to fit the HAADF-STEM image of Ir/CoFe-LDH, which also suggests single atomic dispersion of Ir with different brightness (Supplementary Figs. 6 and 7). The elemental mapping (Fig. 1b) analysis shows homogeneous distribution of Ir along with Cl on the surface of CoFe-LDH. Furthermore, X-ray diffraction (XRD) patterns of the as-prepared CoFe-LDH and Ir/CoFe-LDH display the same Bragg reflections in good agreement with the hexagonal-phase LDH (black line, PDF#40-0215, as shown in Fig. 1c), in accordance with the selected area electron diffraction (SAED) patterns (Supplementary Fig. 8).

To further confirm the atomic structure of Ir, in situ DRIFTS measurements of Ir/CoFe-LDH and Ir$_{cluster}$/CoFe-LDH were performed (Supplementary Fig. 9). After adsorption of CO on Ir/CoFe-LDH, two bands (2080 and 2010 cm$^{-1}$) can be clearly seen in the C−O vibrational frequency region (Supplementary Fig. 9a). These two bands can be assigned to the symmetric (vs) and the anti-symmetric (vas) vibrational

modes of Ir gem-dicarbonyl, Ir(CO)$_2$, respectively, on the basis of reports of atomically dispersed Ir(CO)$_2$ gem-dicarbonyl supported on TiO$_2$ and γ-Al$_2$O$_3$[35]. Of note, dicarbonyl species can only be formed on isolated species[36], thus confirming the atomic dispersion of Ir in Ir/CoFe-LDH. In contrast, Ir$_{cluster}$/CoFe-LDH exhibits multiple overlapped IR peaks between 2000 and 2080 cm$^{-1}$(Supplementary Fig. 9b). The strongest peak at 2046 cm$^{-1}$ is assigned to the linear binding of CO on Ir clusters, which indicates the existence of Ir clusters. However, this atop peak is not observed in Ir/CoFe-LDH, further providing experimental validation for the atomic dispersion of Ir in Ir/CoFe-LDH (Supplementary Fig. 9c). These results suggest the absence of Ir clusters or nanoparticles in Ir/CoFe-LDH, consistent with the HAADF-STEM results.

X-ray photoelectron spectroscopy (XPS) was performed to examine the surface electronic structure and the influence between the CoFe-LDH and the single atomic Ir (Supplementary Fig. 10). The high-resolution Ir 4f XPS spectrum showed two distinct peaks belonging to Ir 4f 7/2 and Ir 4f 5/2: according to the binding energy of the standard samples of IrO$_2$ and H$_2$IrCl$_6$ (Supplementary Fig. 11), the Ir 4f 7/2 at 61.7 eV and 62.4 eV were identified as the Ir-O and Ir-Cl, respectively (the satellites of Ir-O and Ir-Cl were taken into consideration in fitting the XPS spectrum). The high-resolution Cl 2p XPS spectrum can be deconvoluted into 199.1 eV and 198.45 eV, originating from Ir-Cl and Cl$^-$ on LDH[37,38]. Comparing the valence states of Co and Fe in Ir/CoFe-LDH and CoFe-LDH, it was noticed that Co and Fe have obvious shift of binding energy, indicating strong electronic interaction between single atomic Ir and CoFe-LDH.

To further determine the electronic structure and local coordination environment of atomic Ir, X-ray absorption near edge structure (XANES) and extended X-ray absorption fine structure (EXAFS) were performed[39,40]. The Ir L$_3$-edge XANES spectrum (Fig. 1d) indicates that the valence state of Ir in Ir/CoFe-LDH is slightly higher than +4. In the EXAFS spectra (Fig. 1e) as referred to Ir foil, IrCl$_3$, and IrO$_2$, the Ir/CoFe-LDH displays two peaks at 1.62 and 1.99 Å, which can be assigned to Ir-O and Ir-Cl, respectively. Furthermore, the absence of Ir-Ir and Ir-O-Ir bonds suggested atomically distributed Ir. The EXAFS fitting data (Fig. 1f, Supplementary Fig. 12 and Supplementary Table 1) showed Ir-O bonds (1.99 Å) with a coordination number (CN) of 3.1 and Ir-Cl bonds (2.33 Å) with a CN of 2.5 without Ir-Ir or Ir−(O)−Ir contribution in Ir/CoFe-LDH. Additionally, there are backscatter contributions of Ir and M (Co and Fe) at 3.20 Å with a CN of 2.0 at the second shell.

### Electrochemical performance of Ir/CoFe-LDH

The OER performance of the as-prepared Ir/CoFe-LDH was measured in 6.0 M sodium hydroxide (NaOH) aqueous solution using a three-electrode configuration. Supplementary Fig. 13 displays the scanning electron microscopy (SEM) images of the working electrode. The saturated calomel electrode (SCE) (Supplementary Figs. 14 and 15) reference electrode against RHE scale was directly measured by a three-electrode setup, consisting of 2 Pt wires and 1 reference electrode to be calibrated at saturation of H$_2$ (Supplementary Fig. 16). The cyclic voltammetry (CV) curve (Fig. 2a) reveals that the Ir/CoFe-LDH just requires an overpotential of 236 mV to reach a current density of 10 mA cm$^{-2}$, which is 141 mV and 535 mV lower than that of CoFe-LDH and commercial IrO$_2$, respectively. Different from CoFe-LDH, significant reduction in the Co$^{2+}$/Co$^{3+}$ redox peaks are observed after loading single atomic Ir (Supplementary Fig. 17). It suggests that the oxidation peak at 1.24 V versus RHE and the reduction peak at 1.17 versus RHE is a redox couple, which could be assigned to the redox of Ir. And the oxidation peak at 1.33 V versus RHE and the reduction peak at 1.07 vs. RHE is a redox couple, which could be assigned to the redox of Co. Interestingly, when NaCl is added into the electrolyte to mimic the accumulate-to-saturated seawater (6 M NaOH + 2.8 M NaCl)[8], there shows a 34 mV decrease in overpotential from 236 mV to 202 mV to reach a current density of 10 mA cm$^{-2}$ (Fig. 2b), suggesting that

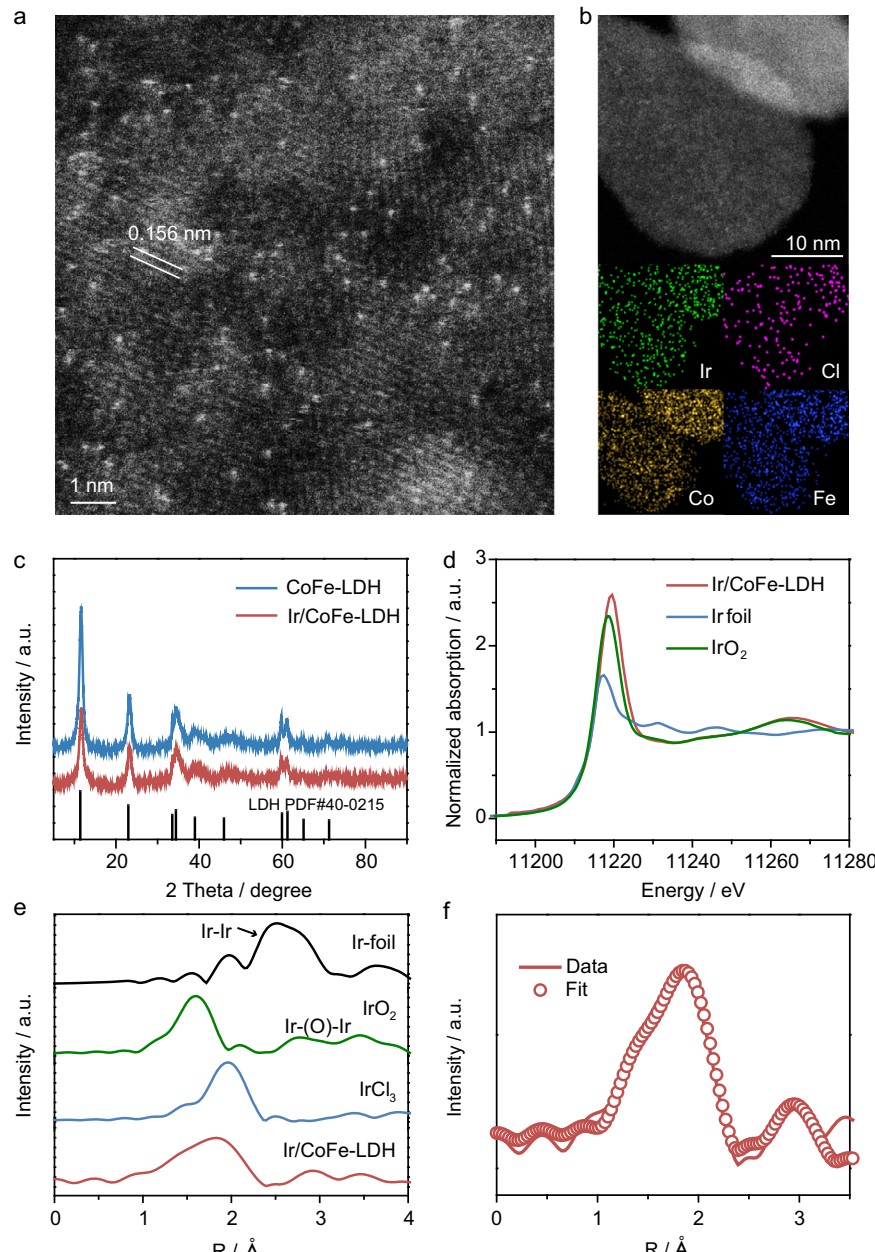

**Fig. 1 | Structural characterizations of Ir/CoFe-LDH. a** HAADF-STEM image of Ir/CoFe-LDH showing single atomic Ir dispersion on CoFe-LDH. **b** Dark-field TEM image of Ir/CoFe-LDH and the corresponding elemental mappings showing the distribution of Fe, Co, Cl, and Ir elements. **c** XRD patterns of the as-prepared CoFe-LDH and Ir/CoFe-LDH. **d** XANES spectra and (**e**) Fourier-transformed Ir $L_3$-edge EXAFS spectra of Ir/CoFe-LDH, Ir foil, IrCl₃ and IrO₂. **f** $k^3$-weighted EXAFS spectrum with fitting (dashed line).

presence of NaCl improve the catalytic performance of Ir/CoFe-LDH. In addition, the turnover frequency (TOF) per Ir-site on Ir/CoFe-LDH (0.76 $O_2$ $s^{-1}$) at the potential of 1.45 V (vs. RHE) in saturated seawater electrolyte is 6.3 times greater than that in NaOH electrolyte (0.12 $O_2$ $s^{-1}$) (Fig. 2b). Taking into consideration the potential issue of excessive catalyst loading at 2 mg/cm², five data points with different catalyst's loading in the range of 0.1–2 mg/cm² were selected to examine the relationship between catalyst's loading and activity (Supplementary Fig. 18a, b). The findings indicate a linear correlation between catalyst's loading and performance within 0.1–0.5 mg/cm² range. Beyond 0.5 mg/cm², the change in catalyst loading shows minimal effect on catalytic performance (reaching a plateau region). Thus, within the linear region, the CV data associated with a catalyst's loading of 0.1 mg/cm² was employed for the precise determination of TOF values. The results

demonstrate that, at a voltage of 1.5 V (vs. RHE), the TOF value per Ir site on Ir/CoFe/LDH in 6 M NaOH + 2.8 M NaCl (7.46 $O_2$ $s^{-1}$) is 7 times higher than that in 6 M NaOH (1.05 $O_2$ $s^{-1}$) (Supplementary Fig. 18c). Meanwhile, it should be noted that the loading amount of Ir should be carefully tailored; insufficient or over-loading would cause less efficient catalysis (Supplementary Figs. 19–21 and Supplementary Tables 2 and 3), which matches to the characteristic of the reported monatomic catalyst. For selectivity, even in saturated seawater electrolyte, Ir/CoFe-LDH displays a close to 100% OER Faradaic efficiency (Supplementary Fig. 22).

The OER performance of Ir/CoFe-LDH exhibits sensitivity to the electrolyte composition with varying NaOH and NaCl ratios. Notably, an optimized [Cl⁻]/[OH⁻] ratio for OER emerges as a significant consideration (Fig. 2c). By altering the NaOH concentration to 1 M, 2 M, 3 M, and 6 M, the OER overpotential at a current density of 10 mA cm⁻²

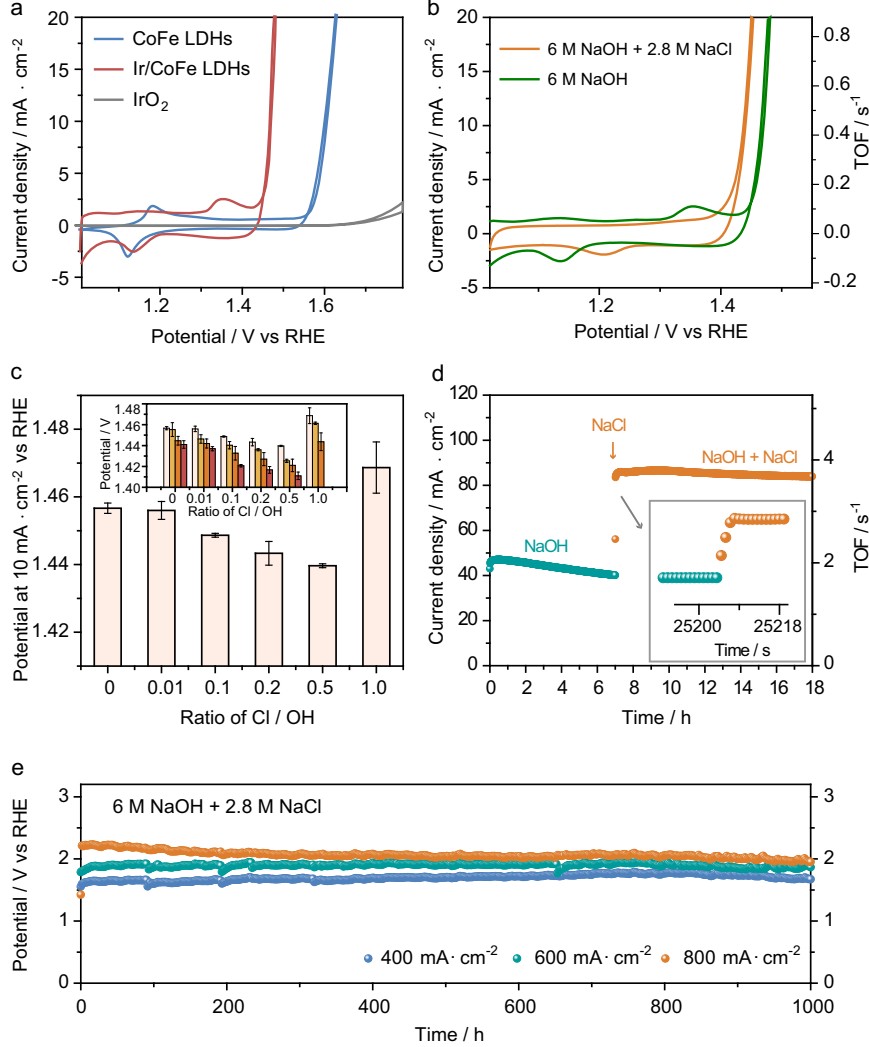

**Fig. 2 | OER performance. a** CV curves of Ir/CoFe-LDH, CoFe-LDH, and $IrO_2$ recorded in 6 M NaOH (the resistance used for iR calibration is 1.53, 1.54, and 1.6 Ω, respectively). **b** Comparison of CV curves and TOF of Ir/CoFe-LDH recorded in 6 M NaOH and 6 M NaOH + 2.8 M NaCl (the resistance used for iR calibration is 1.55, and 1.53 Ω, respectively). **c** Comparison of OER overpotential at a current density of 10 mA cm$^{-2}$ of Ir/CoFe-LDH in electrolyte with different Cl$^-$/OH$^-$ ratios in 1 M NaOH. The inset shows the OER overpotential of Ir/CoFe-LDH at a current density of 10 mA cm$^{-2}$ with different Cl$^-$/OH$^-$ ratios in 1 M (white), 2 M (yellow), 3 M (orange), and 6 M NaOH (red). Three measurements were conducted for each data point with the error bars corresponding to the standard deviation. **d** The change in current density and TOF when the electrolyte was switched from 6 M NaOH to 6 M NaOH + 2.8 M NaCl recorded at 1.48 V versus reversible hydrogen electrode (RHE). **e** The stability test performed in 6 M NaOH + 2.8 M NaCl. (All the catalyst mass-loading was 2 mg/cm$^2$, and the PH of the electrolyte measures 14.78.).

experiences an initial decrease followed by an increase, correlating with the rise in Cl$^-$ concentration. Remarkably, the optimal OER performance consistently appears at the [Cl$^-$]/[OH$^-$] ratio of 1:2 (Supplementary Fig. 23), highlighting the critical impact of Cl$^-$ on OER performance.

In addition, when examined in sole NaOH electrolyte at a voltage of 1.49 V (vs. RHE), the current density of Ir/CoFe-LDH decreased from 47 mA cm$^{-2}$ to 36 mA cm$^{-2}$ in less than 7 h (Fig. 2d). Intriguingly, the incorporation of NaCl into the NaOH electrolyte promptly elevated the current density from 36 mA cm$^{-2}$ to 85 mA cm$^{-2}$ within less than 6 s. Besides, it was noted that Br$^-$ exhibited a similar capability to enhance the OER activity of Ir/CoFe-LDH, while conversely, F$^-$ was found to diminish the catalytic activity (Supplementary Fig. 24). In contrast, other noble metal single atoms introduced onto CoFe-LDH (such as Ru/CoFe-LDH or Rh/CoFe-LDH) did not exhibit such pronounced OER activity enhancement through halogen modification (Supplementary Fig. 25), underscoring the distinctive role played by Ir. This distinctiveness might be attributed to the robust Ir-X (X = Cl, Br or F) interaction, as evident from the UV-visible spectra (Supplementary Fig. 26).

At the same time, no significant OER activity enhancement could be observed in NaOH combined with other salt (e.g., Na$_2$SO$_4$) electrolyte (Supplementary Fig. 27).

To investigate whether SO$_4^{2-}$ would impact the Ir-Cl coordination, we conducted cyclic voltammetry tests on Ir/CoFe-LDH in a solution of 6 M NaOH + 2.8 M NaCl. Subsequently, we gradually introduced 0.03 M Na$_2$SO$_4$ (the sulfate ion concentration in seawater[41]) and continued the CV tests. The results indicate virtually sustained performance (within 3 mV), affirming that SO$_4^{2-}$ does not affect the catalyst's activity (Supplementary Fig. 28). To demonstrate whether SO$_4^{2-}$ under cyclic conditions would displace coordinated Cl$^-$, we conducted a 2-h CV test, the results showed no discernible change in performance before and after the CV cycling (Supplementary Fig. 29). Subsequently, more SO$_4^{2-}$ was introduced into the electrolyte, eventually reaching a concentration of 0.3 M SO$_4^{2-}$ (tenfold enrichment of sulfate concentration in real seawater). CV curves indicate that the catalyst's performance fluctuation is negligible (≤5 mV) within the range SO$_4^{2-}$ concentration of 0.03 M ~ 0.3 M (Supplementary Fig. 28). This suggests that after the formation of a stable Ir-Cl coordination state, the

presence of $SO_4^{2-}$ does not exert any discernible influence on the performance.

Besides activity and selectivity, stability is another crucial consideration for practical application[42]. The deactivation of LDH can be attributed to local pH reduction, interlayer acidification, cation dissolution, or oxidation of metal centers to high-valent cations leading to their leaching[43] and in seawater environment, the presence of chloride ions may exacerbate these risks. The stability of Ir/CoFe-LDH was further accessed in 6 M NaOH + 2.8 M NaCl solution at current densities ranging from 400 to 800 mA cm$^{-2}$ to meet the industry requirements. The applied potentials showed a negligible increase after 1000 h of continuous reaction (Fig. 2e). Meanwhile, performance in real seawater holds significant implications for the application of seawater electrolysis technology. Ir/CoFe-LDH required an overpotential of only 208 mV to achieve a current density of 10 mA cm$^{-2}$ in a 6 M NaOH + seawater electrolyte (Supplementary Fig. 30), indicating its practicability under real industrial conditions. Additionally, CV tests conducted after stability tests at 24, 200, and 400 h confirm the stability of Ir/CoFe-LDH under realistic operating conditions (Supplementary Fig. 31). An electrolyzer with Ir/CoFe-LDH as an anode and NiCoFeP as a cathode at a current density of 1 A/cm$^2$ in a 6 M NaOH + seawater environment was tested. The results illustrate that the electrolyzer can maintain stable performance under high current density for over 2000 h (Supplementary Fig. 32), confirming the viability of Ir/CoFe-LDH catalyst for real seawater electrolysis.

Furthermore, additional characterizations of post-reaction catalysts were performed to confirm the stability of Ir/CoFe-LDH. SEM after long-term stability test revealed no significant changes in the morphology of the nanosheets (Supplementary Fig. 33), while TEM after long-term stability test showed absence of clusters or particles (Supplementary Fig. 34). HADDF-STEM characterizations (Supplementary Fig. 35) show that the distribution of atomic Ir atoms on CoFe-LDH surface after long-term stability test has no obvious change. There is still no Ir-Ir or Ir−O−Ir signal in the EXAFS data of post-OER catalyst, also revealing isolated dispersion of Ir atoms after long-term OER stability tests. The XRD pattern of the post-OER Ir/CoFe-LDH (Supplementary Fig. 36) displays the characteristic peak of LDH at about 11.6°. The XPS measurement of Ir/CoFe-LDH after long-term OER stability test (Supplementary Fig. 37) revealed that the oxidation state of Co and Fe showed no significant change, confirming that Ir-Cl coordination could stabilize CoFe-LDH, preventing oxidation and dissolution of CoFe-LDH, thereby ensuring the stability of CoFe-LDH substrate. Furthermore, XPS quantitative analysis indicated a negligible change of the surface Ir concentration before and after OER. The dissolved Ir in the electrolyte after OER stability test was also quantified by inductively coupled plasma optical emission spectroscopy (ICP-MS, 9.558 ppb), which is nearly nine times less than the physical mixture of IrO$_2$ and CoFe-LDH (82.308 ppb), as shown in Supplementary Table 4. This experimental evidence further substantiates the stability of Ir loading on CoFe-LDH. Ir stabilizes CoFe through controllable Ir-Cl coordination, suppressing CoFe oxidation, in the meanwhile CoFe also stabilizes Ir through strong electronic interaction. All of the above characterizations together with the electrochemical data suggest good stability of Ir/CoFe-LDH in seawater electrolysis. Additionally, it should be noted that at current densities of 600 and 800 mA cm$^{-2}$, the overpotential was thermodynamically high enough to trigger the chloride oxidation reaction (thermodynamic potential: $2Cl^- \rightarrow Cl_2 + 2e^-$ E = 1.33 V versus RHE, $Cl^- + 2OH^- \rightarrow OCl^- + H_2O + 2e^-$ E = 1.66 V versus RHE at 2.8 M NaCl), but no chlorine or hypochlorite was detected (with oxygen Faradaic efficiency > 99.9935% as shown in Supplementary Figs. 38–40), suggesting a high OER selectivity for Ir/CoFe-LDH in high concentration NaCl electrolyte.

## In situ XAS and Raman characterization of Ir/CoFe-LDH

To explore why and how Cl$^-$ can boost the OER activity of Ir/CoFe-LDH, *operando* EXAFS at the Ir L$_3$-edge was performed and compared in NaOH + NaCl (Fig. 3a) and NaOH (Fig. 3b) electrolyte. The local coordination of the two Ir/CoFe-LDH samples used for *operando* EXAFS are nearly the same, implying that the differences of EXAFS spectra in NaOH + NaCl and NaOH electrolyte arise from the catalyst's structure evolution during OER (Supplementary Fig. 41 and Supplementary Table 5). Subsequently, under open circuit voltage (OCV), the coordination of both catalysts remains nearly unaltered (Supplementary Fig. 42 and Supplementary Table 6). However, at a potential of 1.57 V versus RHE, only Ir-O coordination was evident in the EXAFS spectrum of Ir/CoFe-LDH immersed in NaOH (Supplementary Fig. 43 and Supplementary Table 7), suggesting that OH$^-$ formed a competitive adsorption for Ir-Cl coordination. In contrast, both Ir-Cl and Ir-O could be observed at the same applied potential in NaOH + NaCl electrolyte. This observation can be attributed to the substantial Cl$^-$ concentration, allowing for the persistence of Ir-Cl coordination. The above results suggest a competitive binding between Cl$^-$ and OH$^-$ with single atomic Ir sites on CoFe-LDH during OER in the seawater environment, and the dynamic stability of Ir-Cl coordination in seawater may stand as a key factor driving the OER performance enhancement. Concurrently, a distinctive peak at ~2.6 Å was detected in the EXAFS spectra, which could be attributed to the close proximity of Ir and M (M = Co or Fe) due to the oxidation of CoFe-LDH surface to MOOH during the OER process.

In situ Raman spectroscopy was further performed on Ir/CoFe-LDH to obtain mechanistic insight into the influence of Cl$^-$ adsorption over single Ir atomic site on the OER activity. Compared to CoFe-LDH (Supplementary Fig. 44), the Raman spectra of Ir/CoFe-LDH at OCV showed a new peak at around 333 cm$^{-1}$ (Fig. 3c−f). This peak can be assigned to Ir-Cl vibration according to the standard sample IrCl$_3$, as shown in Supplementary Fig. 45.

Moreover, In NaOH + NaCl environment (Fig. 3c, d), the presence of Ir-Cl coordination at 333 cm$^{-1}$ was consistently observed. In contrast, in alkaline electrolytes without Cl addition, Ir-Cl coordination was only observed under open circuit voltage (OCV) condition, and its presence ceased upon application of voltage (Fig. 3e, f), indicating its dynamic nature. This phenomenon suggests that under high OH$^-$ concentration, there is a pronounced competition for adsorption between Cl$^-$ and OH$^-$, necessitating a high NaCl concentration to uphold the stability of the coordination. This observation is consistent with our XAS data, where the coordination between Ir and Cl remains intact in high salt concentration environment, whereas it dissipates in an OH$^-$-rich environment. Consequently, it becomes essential to maintain a relatively high salt concentration environment to uphold the Ir-Cl coordination, thereby facilitating the kinetics of OER.

**Theoretical investigation.** We performed computational studies aimed at identifying the effects of Cl$^-$ modulation of the local iridium coordination environment and the resulting impacts on the electronic structure of iridium center as well as the energetics of OER[44–46]. Based on the EXAFS fitting results and the characterization of catalyst after OER, the catalyst was found to undergo reconstruction during OER and the real active structure is the single-atom iridium bound on the surface of oxyhydroxide (i.e., MOOH, M = Co/Fe), and the reconstruction of hydroxide to oxyhydroxide during OER has been observed in the literature[47,48]. Hence, we built our simulation models as shown in Supplementary Fig. 46. Penta-coordinated Ir with three states: Ir coordinated with two OH (Ir$_{-OH,OH}$), with one Cl and one OH (Ir$_{-OH,Cl}$), and with two Cl (Ir$_{-Cl,Cl}$) anchored on the surface of two-layer periodic LDH, were constructed as the simulation models. For pure CoFeOOH, unsaturated Fe sites in the (100) surface were used as the active sites. Thermodynamic analyses (Fig. 4a and Supplementary Fig. 47) suggest that the formation of *OOH intermediate is the rate-determining step

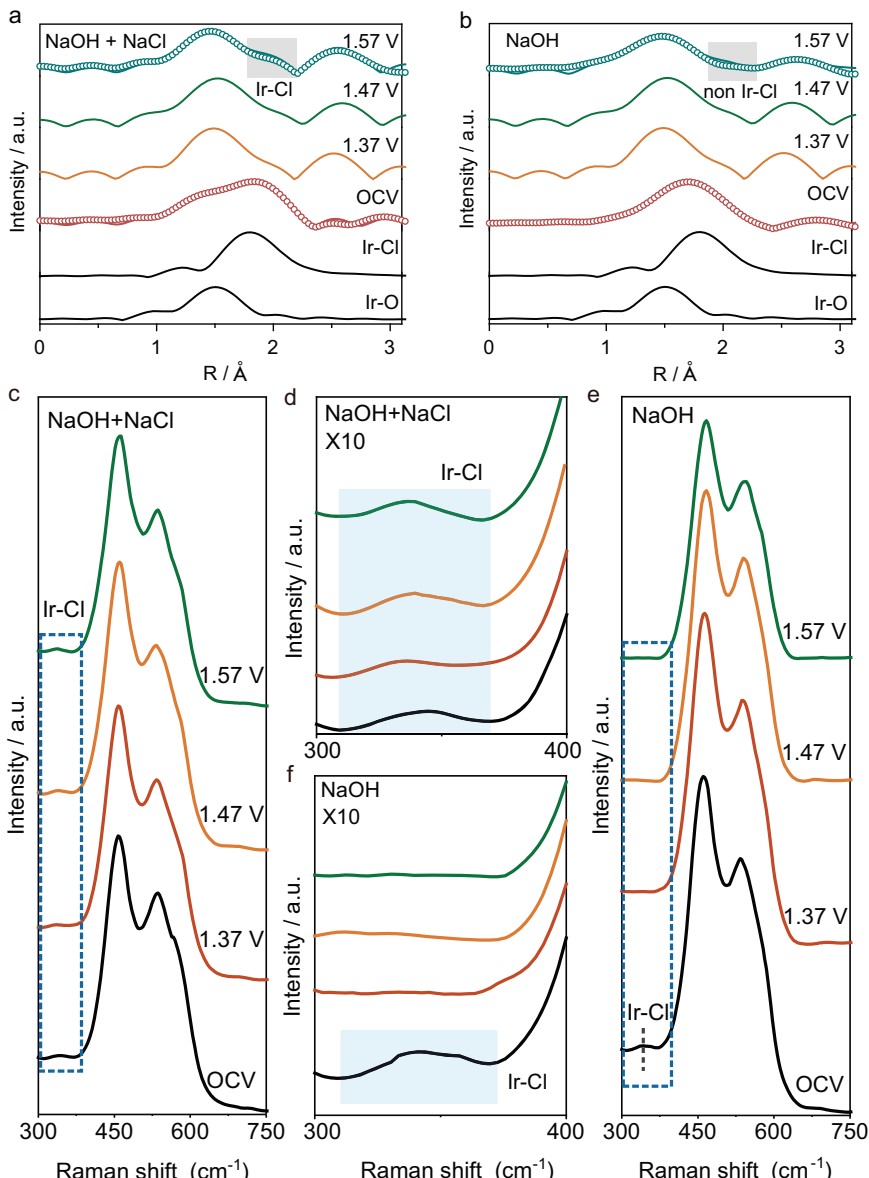

**Fig. 3 | In situ XAS and Raman measurement. a** In situ Fourier-transformed EXAFS spectra of Ir/CoFe-LDH in (**a**) NaOH + NaCl and (**b**) NaOH. In situ Raman spectra of Ir/CoFe-LDH recorded in (**c**), (**d**) NaOH + NaCl, and (**e**), (**f**) NaOH. The blue-highlighted portions correspond to the Ir-Cl coordination.

(RDS) of OER on Fe sites in CoFeOOH and Ir centers in Ir/CoFeOOH. The single Ir atom with 2 OHs (Ir$_{-OH,OH}$) could lower the *O/*OOH oxidation free energy from 2.24 eV to 1.73 eV. Mono chloride coordination with Ir (Ir$_{-OH,Cl}$) could further lower this free energy to 1.51 eV in preference to Ir$_{-OH,OH}$ and meanwhile enhance the *OH adsorption, which is the key step competing with chloride adsorption and oxidation. However, further coordination with another chloride ion to form two Ir-Cl bonds would increase the free energy of *O/*OOH oxidation to 1.64 eV. The activity sequence of the coordinated Ir species follows Ir$_{-Cl,OH}$ > Ir$_{-Cl,Cl}$ > Ir$_{-OH,OH}$. Figure 4b displays the two-dimensional (2D) map of theoretical OER onset overpotential ($\eta$) for single atomic Ir anchored on CoFeOOH with different coordination states by assuming the scaling relationship of $\Delta G_{OOH} = 0.89\Delta G_{OH} + 3.12$ (Supplementary Fig. 47c). Other halogens like Br$^-$ and F$^-$ could also affect the adsorption energy of *O and *OH as well as the overpotential of OER, but was not as efficient as Cl$^-$ (Supplementary Fig. 47b) in accordance with the experimental results.

To further understand the underlying OER mechanism and the role of Cl, projected crystal orbital Hamiltonian population (pCOHP)

analysis between the Ir centers and the O atoms in the adsorbed *OOH were calculated to give information on OOH intermediates adsorption[49,50]. We calculated the integrated COHP (ICOHP) up to the highest occupied bands (below the E$_f$), which directly gave quantitative information on the bonding states. In general, a negative −COHP indicates an antibonding state and a positive −COHP indicates a bonding state[51,52]. As shown in Fig. 4c, the lower ICOHP value (−4.60 eV) demonstrates that the molecular orbitals of adsorbed OOH interact much stronger with the Ir bands in Ir$_{-Cl,OH}$ than in IrO$_2$ (−3.89 eV), Ir$_{-OH,OH}$ (−4.23 eV) and Ir$_{-Cl,Cl}$ (−4.27 eV), indicating stronger OOH adsorption on Ir$_{-Cl,OH}$. These results suggest that the *OOH formation, which is the RDS of OER, is easier to take place on the Ir$_{-Cl,OH}$ center, thereby promoting the OER kinetics.

To get experimental evidence on the formation of *OOH intermediate, electron paramagnetic resonance (EPR) spectroscopy test was performed using 5,5-dimethyl-1-pyrroline N-oxide (DMPO) as an *OOH free radical trapping agent[53,54]. As shown in Fig. 4d, the signal originating from *OOH recorded in NaOH + NaCl electrolyte is nearly twice as much as that recorded in NaOH electrolyte, matching well with

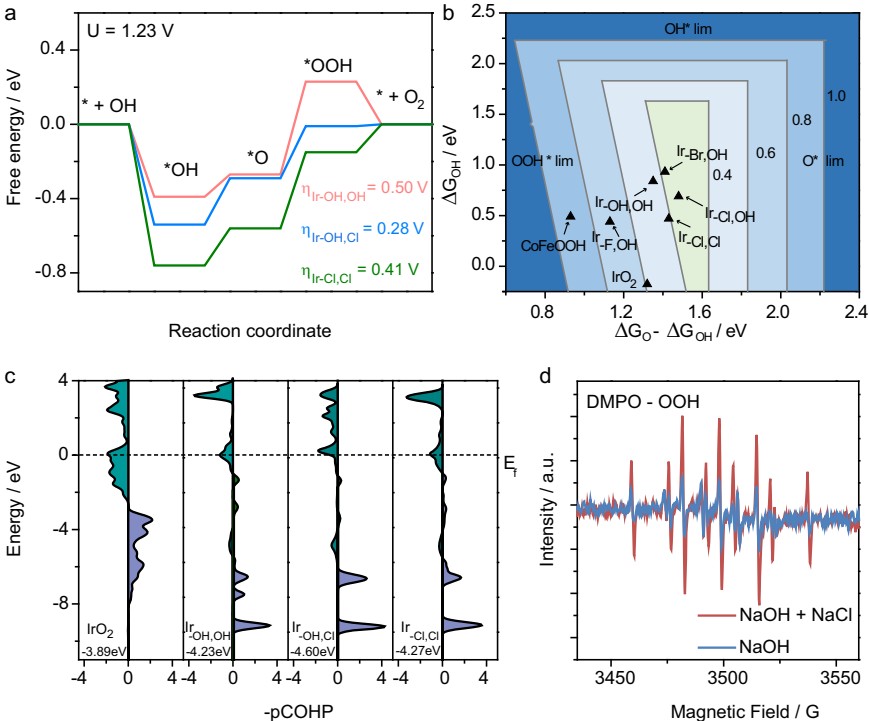

**Fig. 4 | Theoretical investigation. a** The activation energy of Ir/CoFe-LDH with three coordination states. **b** 2D map of theoretical OER onset overpotential ($\eta$), the contour map is constructed by assuming the scaling relationship of $\triangle G_{OOH} = \triangle G_{OH} + 3.17$. **c** Projected crystal orbital Hamiltonian population (pCOHP) between the Ir center and the O atom in *OOH. **d** EPR detection of DMPO-OOH. The sample was taken after 10 min of electrolysis at 1.42 V versus RHE.

the COHP data. The enhanced adsorption of OER intermediates can be further deduced from the smaller slope of the open circuit potential (OCP) decay collected in NaOH + NaCl electrolyte as compared to that collected in pure NaOH electrolyte, as shown in Supplementary Fig. 48.

To further verify the specific role of Ir-Cl, we conducted calculations of the Density of States (DOS) and Projected Density of States (PDOS) for Ir-Cl/CoFe-LDH and Cl/CoFe-LDH. The results indicate that Cl hybridizes with the conduction bands of Co and Fe, but there is no hybridization with the valence bands (Supplementary Fig. 49b). Upon loading a Ir single-atom, Ir-Cl exhibits significantly stronger hybridization with both the valence and conduction bands (Supplementary Fig. 49a), highlighting the enhanced orbital hybridization and bonding capability of Ir-Cl. The total density of states (TDOS) data aligns with the partial density of states (PDOS) data, demonstrating a pronounced increase in valence band hybridization in the presence of surface Ir, in contrast to CoFe-Cl case where the valence bands show no interaction (Supplementary Fig. 50).

To rationalize the improved OER performance and explore the role of chloridion, kinetic characteristics of Ir/CoFe-LDH in NaOH + NaCl and NaOH electrolyte were systematically studied[55]. The activation energy for OER over the Ir/CoFe-LDH catalyst in NaOH + NaCl electrolyte was estimated to be ~34.21 kJ/mol using the Arrhenius equation, much lower than that in NaOH electrolyte (~66.15 kJ/mol), verifying the much-improved OER kinetics by Cl⁻ adsorption (Supplementary Figs. 51 and 52). On the other hand, ClOR displays remarkably high theoretical overpotentials (Supplementary Fig. 53), offering a compelling rationale for the enduring near 100% OER selectivity of Ir/CoFe-LDH at industrial current densities (400–800 mA cm⁻²) in seawater electrolysis.

To further demonstrate the stability of Ir/CoFe-LDH, we selected Cl adsorption sites on Co or Fe, respectively. The results indicate that, in the presence of Ir, Cl cannot effectively adsorb onto Co (Supplementary Movie 1) or Fe (Supplementary Movie 2), respectively

(Supplementary Fig. 54). The optimization process revealed direct Cl adsorption onto Ir sites, eliminating the existence of stable configurations like Co-Cl or Fe-Cl (Supplementary Figs. 55 and 56). Consequently, these two active sites do not exist at presence of Ir. This finding further confirms the significant improvement in the electrochemical stability of CoFe LDH upon the introduction of Ir. The strong Ir-Cl coordination stabilizes the CoFe sites, inhibiting Cl coordination with both metal sites, thus ensuring system stability in seawater electrolysis.

In summary, we have devised an ingenious strategy involving the deposition of single-atom Ir onto CoFe-LDH and leveraging abundant Cl⁻ ions within the seawater milieu to dynamically regulate the coordination state of Ir single-atom catalyst. This orchestration has endowed Ir/CoFe-LDH catalyst with greatly enhanced OER reactivity in seawater, surpassing its performance in alkaline environment. The confluence of *operando* characterizations and DFT calculations substantiates that the amplified OER performance of Ir/CoFe-LDH in seawater originates from the dynamic regulation of the Cl and OH coordination states on Ir. This chloride-mediated coordination augmentation facilitates robust adsorption of *OOH intermediates in OER, thereby reducing the activation energy barrier for the rate-determining OOH* formation by a factor of 1.93 and significantly increasing the cathodic interfacial electron transfer rate (CIER). Impressively, Ir/CoFe-LDH exhibits a remarkable operational stability at industrial current densities (400–800 mA cm⁻²) in seawater, maintaining uninterrupted activity for over 1000 h while retaining a remarkable 99.99% selectivity. This study not only unveils the immense potential of Ir/CoFe-LDH in seawater electrolysis, but also presents a novel strategy for regulating single-atom Cl coordination in seawater environments, a breakthrough that holds the promise of minimizing energy consumption and costs associated with seawater electrolysis, thereby propelling the practical deployment of this technology.

# Methods

## Materials

Iron(III) nitrate nine-hydrate ($Fe(NO_3)_3 \cdot 9H_2O$, 99.99%), cobalt(II) nitrate hexahydrate ($Co(NO_3)_2 \cdot 6H_2O$, 99.99%), iridium chloride hydrate ($IrCl_3 \cdot xH_2O$, 99.9%), potassium chloride (KCl, ≥99%) were purchased from Sigma-Aldrich. Sodium hydroxide (NaOH, ≥96%), sodium chloride (NaCl, ≥99.5%), sodium carbonate ($Na_2CO_3$, ≥99.5%) were purchased from Fuchen Chemical Reagent Co., Ltd. Ethanol ($CH_3CH_2OH$, ≥99.5%) was purchased from Tianjin Fuyu Fine Chemical Co., Ltd. 5,5-Dimethyl-1-pyrroline N-oxide (DMPO) was purchased from Dojindo China Beijing Co., Ltd. Carbon fiber paper (Toray 060) was purchased from Suzhou Sinero Technology Co., Ltd. Seawater was taken from the Yellow Sea, China. Deionized (DI) water (resistivity: 18.3 MΩ cm) was used for the preparation of all aqueous solutions.

## Preparation of CoFe-LDH

CoFe-LDH was synthesized via a co-precipitation method. In brief, 40 ml solution A: $Fe(NO_3)_3 \cdot 6H_2O$ (0.05 M) and $Co(NO_3)_2 \cdot 6H_2O$ (0.1 M) and 40 ml solution B: NaOH (0.75 M) and $Na_2CO_3$ (0.125 M) were firstly prepared. Solution A and B were added dropwisely at the same time into a beaker filled with 40 ml of deionized water. The pH value of the final suspension was adjusted to 8.5 under magnetic stirring, as monitored using a pH meter at room temperature (25 °C). The pH is well controlled since salt solution A and alkali solution B was added at the same time, and the ratio of these two solutions could be adjusted to meet the targeting pH. The resulted suspension was aged for 12 h at 25 °C. Subsequently, the precipitate was collected by centrifugation, and washed 3 times using DI water, followed by ethanol wash for 3 times. The final precipitates were dried in a vacuum oven at 60 °C overnight.

## Preparation of Ir/CoFe-LDH

The as-prepared CoFe-LDH (0.2 g) was added to 50 ml DI water under magnetic stirring to form a colloid suspension. Then, 10 ml freshly made aqueous $IrCl_3$ solution (5 mg) containing 0.02 M NaOH was dropwisely added to the above CoFe-LDH colloid suspension. The suspension was stirred with simultaneous heating at 60 °C for 6 h. Afterwards, the precipitate was collected by centrifugation, washed by DI water and ethanol, each for 3 times. Then, the final precipitates were dried in a vacuum oven at 60 °C overnight.

## Preparation of Ir$_{cluster}$/CoFe-LDH

The as-prepared CoFe-LDH (0.2 g) was added to 50 ml DI water under magnetic stirring to form a colloid suspension. Then, 10 ml freshly made aqueous $IrCl_3$ solution (10 mg) containing 0.02 M NaOH was dropwisely added to the above CoFe-LDH colloid suspension. The suspension was stirred with simultaneous heating at 60 °C for 6 h. Afterwards, the precipitate was collected by centrifugation, washed by DI water and ethanol each for 3 times. Then, the final precipitates were dried in a vacuum oven at 60 °C overnight.

## HAADF-STEM characterization and simulation

Before imaging, the as-prepared catalysts were added into anhydrous ethanol by using an ultrasonator to form a very dilute colloid suspension, then 20 μl suspension was dripped onto 230 mesh Cu grids coated with ultrathin carbon. The high-resolution HAADF-STEM image was acquired using a Thermo Fisher Spectra 300 microscope equipped with an aberration corrector for the probe-forming lens, operated at 300 kV. The beam current was lower than 40 pA and the STEM convergence semi-angle was ~25 mrad, which provided a probe size of ~0.6 Å at 300 kV. The HADDF-STEM images with a 30° tilt of the sample were taken by tilting double tilt holder −30°(α), which can be moved in α (±35°) and β (±30°) directions. The HADDF-STEM image simulations were carried out using Dr. Probe software[34], the parameters were set as

same as the experimental condition. The accuracy of simulation results was 0.008 nm/pix.

## Characterization instruments

Transmission electron microscopy measurement was carried out on a JEOL JEM 2100. X-ray powder diffraction (XRD) patterns were recorded on an X-ray diffractometer (Rigaku D/max 2500) with Cu Kα radiation (40 kV, 30 mA, $\lambda = 1.5418$ Å) at a scan rate of 5° min$^{-1}$ in the $2\theta$ range from 3 to 90°. X-ray photoelectron spectroscopy (XPS) spectra were recorded on an ESCALAB 250 (Thermo Fisher Scientific USA) photoelectron spectrometer using monochromate Al Kα 150 W X-ray beam. All binding energies were referenced to the C 1s peak (284.8 eV). Specifically, by evaluating the deviation of the binding energy position corresponding to the C 1s peak at 284.8 eV, the obtained difference is then utilized to carry out a comprehensive calibration of the entire dataset. ICP-MS measurement was performed on a Thermo X Series II ICP-MS quadrupole system, Thermo Fisher Scientific to quantify the chemical composition of the catalyst EPR spectroscopy measurement was performed on a Magnettech MS-5000X using 5,5-dimethyl-1-pyrroline N-oxide (DMPO) as the radical trap.

## X-ray absorption spectroscopy

The measurements were performed in a typical three-electrode setup in a specially designed Teflon container with a window sealed by Kepton tape. The testing conditions were the same as in the electrochemical characterization case. The measurements were performed at BL-17C at the National Synchrotron Radiation Research Center (NSRRC, Hsinchu, Taiwan). In situ data were collected in total-fluorescence mode using a silicon drift detector.

## XANES and EXAFS data analysis

The X-ray absorption near edge structure (XANES) and extended X-ray absorption fine structure (EXAFS) data were analyzed by the software Athena of the IFEFFIT package. The EXAFS spectra were analyzed through post-edge background subtraction from the overall absorption and normalized with respect to the edge-jump step by the software Artemis following previously reported data fitting methodology[56–58].

## In situ Raman test

In situ Raman spectra were collected on LabRAM ARAMIS (Horiba Jobin Yvon, France) with a PSU-H-FDA 532 nm laser source (Changchun New Industries Optoelectronics Tech. Co. Ltd, China). An LMPlanFLN 50× microscope objective lens with a numerical aperture of 0.5 (OLYMPUS, Japan) was used for Raman microscopy. Raman frequency was calibrated by a Si wafer (520.8 cm$^{-1}$) during each experiment. In situ electrochemical Raman experiments were performed in an in situ Raman cell (Tianjin Aida Hengsheng Technology Development Co., Ltd, China) and the prepared catalyst, Hg/HgO, and Pt wire were employed as the working, reference, and counter electrode, respectively. A CHI 660e (Shanghai Chenhua Instrument Co., Ltd, China) electrochemical workstation was used to control the potential where the applied potentials were increased step by step from open circuit potential (OCP) to 1.57 V vs. RHE. Each spectrum was obtained at least three times with an exposure time of 30 s.

## In situ DRIFTS experiments

In situ DRIFTS was used to characterize the interaction of the catalyst with CO. The in situ DRIFTS experiments were performed on a Bruker INVENIO R Fourier transform infrared spectrometer equipped with an MCT/A detector cooled by liquid nitrogen and a Harrick diffuse-reflectance attachment. Approximately 50 mg catalyst was loaded in the Harrick Praying Mantis high-temperature reaction chamber equipped with KBr windows (HVC-DRP-5). The chamber was sealed and gases were flown through at atmospheric pressure. High-purity Ar

was used as the purging gas, and dilute CO (10% CO-certified grade) was used as the probe gas. The temperature was controlled by a thermocouple in direct contact with the sample. Circulating water was used to cool the body of the reaction chamber (set temperature: 22 °C). All of the in situ characterizations followed the same pretreatment procedure, each reported spectrum is an average of 64 scans. A spectral resolution of 4 cm$^{-1}$ was used to collect the spectra, which are reported in Kubelka–Munk units. For Ir/CoFe-LDH, the sample was calcined in Ar (99.999%, 30 ml min$^{-1}$, 0.2 ml min$^{-1}$ s$^{-1}$ ramp rate) at 105 °C (5 °C min$^{-1}$ ramp rate) for 30 min to remove physically adsorbed H$_2$O. The temperature was then cooled in Ar to 25 °C with circulating water and the background spectrum was recorded. The sample was then labeled in CO (10% CO-certified grade, 30 ml min$^{-1}$, 0.2 ml min$^{-1}$ s$^{-1}$ ramp rate) at 25 °C for 0.5 h. Afterwards, Ar (99.999%, 30 ml min$^{-1}$, 0.2 ml min$^{-1}$ s$^{-1}$ ramp rate) was purged at 25 °C for 30 min to remove the physical adsorbed CO until there was no change observed in the spectra. The DRIFTS spectra were recorded in the wavenumber range from 1000 to 4000 cm$^{-1}$ at 298 K with a total of 1000 measurements taken, each separated by a 10-s interval.

## Electrochemical measurements

All electrochemical measurements were performed on a three-electrode setup using a CHI 660e electrochemical workstation. A platinum wire and a SCE double salt bridge electrode were served as the counter electrode and the reference electrode, respectively. The SCE double salt bridge, which was purchased from Tianjin Aida Hengsheng Technology Development Co., Ltd, consists of saturated KCl. The working electrode was prepared by the following steps: (1) Preparation of catalyst ink. 2 mg of the as-synthesized catalyst, 1 mg of carbon black, and 10 μl of Nafion solution were dispersed in ethanol (0.7 ml) and deionized water (0.7 ml) under ultrasonication for at least 1 h to form a homogeneous catalyst ink. (2) Coating the catalyst ink onto carbon fiber paper (1 cm × 1 cm, Toray, TGP-H-060): the catalyst ink was drop-casted onto carbon fiber paper (1 cm × 1 cm, Toray, TGP-H-060) 50 μl each time with drying under an infrared lamp, followed by dropping another 50 μl ink until the catalyst mass-loading of the catalyst reached 2 mg/cm$^2$. Cyclic voltammetry (CV) was measured from 0 to 1.0 V versus SCE at a scan rate of 2 mV s$^{-1}$. The electrochemical impedance spectroscopy (EIS) was measured by applying an AC voltage of 5 mV at the overpotential of 10 mV with a frequency from 100 kHz to 0.1 Hz. All polarization curves were corrected for Ohmic-drop compensation with Ohmic resistance obtained by the EIS. The SCE reference electrode against RHE scale was directly measured by a three-electrode setup, consisting of 2 Pt wires and 1 reference electrode to be calibrated. The open circuit potential (OCP) was measured under the condition of purging high-purity H$_2$ gas into the electrolyte. An OCP was applied after saturation of H$_2$ gas, obtaining a stable potential. This potential is the potential difference between RHE and the reference electrode. In the choice of electrolyte for testing, the use of 6 M NaOH mirrors the alkalinity levels typically employed in industrial alkaline water electrolysis. Moreover, according to our previous work[8], it is notable that the maximal solubility of NaCl in 6 M NaOH is 2.8 M, as excess NaCl concentration beyond this threshold results in crystalline precipitation. Importantly, in practical scenarios, electrolyte salt concentrations are unlikely to exceed this limit. By conducting tests under these conditions, we aim to showcase the electrode's performance at its utmost capacity.

## Measurement of activation energy (E$_a$)

The activation energy of OER is an indicator of reaction kinetics, which is only dependent on the material. The Eq. (1) is Arrhenius equation, where $A$ is the preexponential factor, $R$ is the ideal gas constant, $T$ is the temperature, and $k$ is a rate constant, which can be applied to determine the activation energy of OER on the catalysts. At a fixed overpotential, the kinetic current ($I_k$) has a linear relationship with a rate

constant ($k$), as shown in Eq. (2), where $F$ is Faraday constant, $n$ is the number of transferred electrons, $S$ is the electrode area and $C^*$ is concentration. If we assume that mass transport effects are negligible on the OER currents, the $I_k$ can be expressed as Eq. (3), where $W$ is the activation barrier at a constant potential. The kinetic activation energy at a given voltage can be expressed as Eq. (4). Hence, the activation energy for OER can be defined by Eq. (5):

$$k = Ae^{-Ea/RT} \tag{1}$$

$$I_K = nFSkC^* \tag{2}$$

$$I_K = Ae^{-W/RT} \tag{3}$$

$$\frac{\partial \ln I_K}{\partial 1/T} = -\frac{W}{T} \tag{4}$$

$$W = E_a - \beta\eta \tag{5}$$

## Theoretical methods

All theoretical calculations were performed using the projector-augmented wave method and a plane-wave basis set as implementation in the Vienna Ab Initio Simulation Package (VASP). The bulk and surface properties of CoFeOOH were optimized within GGA-PBE. Full optimization of all atom positions in the bulk was performed via the action of a conjugated gradient optimization procedure. The Monkhorst-Pack k-point samplings were set as 3 × 3 × 1 for the geometry optimization, and 13 × 13 × 1 for the computation of electronic structure. And the bulk constants were optimized using the 3 × 3 × 3 Monkhorst-Pack k-point sampling. The cutoff energy for plane-wave basis functions was set to 600 eV with the energy change convergence criterion of 1 × 10$^{-5}$ eV. Atomic positions were allowed to relax until the sum of the absolute forces reached down to 0.02 eV Å$^{-1}$. Hubbard-U correction method was applied to improve the description of localized Co and Fe d-electrons in the CoFeOOH with U = 5.3 and U = 3.5 for Fe and Co, respectively. The spin polarization, long-range van der Waals interaction (IVDW = 11), and solvent corrections were also included in surface calculations. The solvent effect on adsorbates was considered using the Poisson–Boltzmann implicit solvation model with a dielectric constant of 78.4. The Gibb's free energies were calculated by:

$$\Delta G = \Delta E + \Delta ZPE - T\Delta S$$

where the symbols represent the binding energy ($\Delta E$), the change in zero-point energy ($\Delta ZPE$), temperature ($T$), and the entropy change ($\Delta S$) of the system, respectively. The crystal orbital Hamiltonian population (COHP) was calculated by LOBSTER, and the convergence threshold for the iteration in self-consistent field was set at 1 × 10$^{-8}$ eV.

## Data availability

Source data are provided with this paper.

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

## Acknowledgements

The authors thank the help from Dr. Cejun Hu for the help on characterization. X.S. and Y.K. acknowledge financial support from the National Key Research and Development Project (2021YFA1502200), the National Natural Science Foundation of China (21935001), Beijing Natural Science Foundation (Z210016), a long-term subsidy from China's Ministry of Finance and the Ministry of Education. B.L. acknowledges financial support from the City University of Hong Kong start up fund (9020003) and ITF-RTH-Global STEM Professorship (9446006).

## Author contributions

X.S., B.L., and Y.K. supervised the project. X.D. conceived the idea and carried out the experiments. X.D., Q.S., P.L., and D.Z. conducted material synthesis and electrochemical measurements. Q.S. conducted the in situ DRIFTs and in situ Raman measurements. T.L., G.Y., and W.L. help with the material characterization. W.C., Y.C., and L.Z. conducted the in situ XAS measurements. X.D., Y.L. and E.Y. performed the DFT calculations. H.Y., G.Z., and W.L. assisted with the data analysis. X.S., B.L., Y.K., X.D., and Q.S. wrote the paper. S.H. conducted the in situ XAS measurements in the revised version. J.L. and J.D. analyzed the data of XAS in the revised version. Z.W. and H.X. help with the DFT simulation. J.F. and Z.Z. conducted the ATR-SEIRAS measurements in the revised version. C.J., J.L., and L.B. conducted the HADDF-STEM measurements and simulations. All authors discussed the results and assisted the manuscript preparation.

## Competing interests

The authors declare no competing interests.

## Additional information

¹State Key Laboratory of Chemical Resource Engineering, Beijing Advanced Innovation Center for Soft Matter Science and Engineering, College of Chemistry, Beijing University of Chemical Technology, Beijing 100029, PR China. ²School of Chemistry, Chemical Engineering and Biotechnology, Nanyang Technological University, Singapore 637459, Singapore. ³Beijing National Laboratory for Molecular Sciences, CAS Key Laboratory of Colloid, Interface and Chemical Thermodynamics, Institute of Chemistry, Chinese Academy of Sciences, Beijing 100190, PR China. ⁴Ocean Hydrogen Energy R&D Center, Research Institute of Tsinghua University in Shenzhen, Shenzhen 518057, PR China. ⁵State Key Lab of Organic–Inorganic Composites, College of Chemical Engineering, Beijing University of Chemical Technology, 100029 Beijing, PR China. ⁶Energy & Catalysis Center, School of Materials Science & Engineering, Beijing Institute of Technology, Beijing 100081, PR China. ⁷Hefei National Research Center for Physical Sciences at the Microscale, University of Science and Technology of China, Hefei 230026, PR China. ⁸Beijing Synchrotron Radiation Facility, Institute of High Energy Physics, Chinese Academy of Sciences, 100049 Beijing, PR China. ⁹Department of Chemistry, Tsinghua University, 100084 Beijing, PR China. ¹⁰Department of Materials Science and Engineering, City University of Hong Kong, Hong Kong SAR 999077, PR China. ¹¹College of Energy, Shandong University of Science and Technology, Tsingtao 266590, PR China. ¹²Beijing Key Laboratory of Energy Environmental Catalysis, Beijing University of Chemical Technology, 100029 Beijing, PR China. ¹³Department of Applied Chemistry and Center for Emergent Functional Matter Science, National Yang Ming Chiao Tung University, Hsinchu 300, Taiwan. ¹⁴School of Materials Science and Engineering, Tianjin Key Lab of Photoelectric Materials & Devices, Tianjin University of Technology, Tianjin 300384, PR China. ¹⁵ShenSi Lab, Shenzhen Institute for Advanced Study, University of Electronic Science and Technology of China, Shenzhen 518110, PR China. ¹⁶CAS Key Laboratory of Standardization and Measurement for Nanotechnology, National Center for Nanoscience and Technology, 100190 Beijing, PR China. ¹⁷Department of Chemistry & Center of Super-Diamond and Advanced Films (COSDAF), City University of Hong Kong, Hong Kong SAR 999077, PR China. ¹⁸These authors contributed equally: Xinxuan Duan, Qihao Sha. ✉e-mail: kuangy@tsinghua-sz.org; bliu48@cityu.edu.hk; sunxm@mail.buct.edu.cn

