## [Peer Review File · Nature Communications]

REVIEWER COMMENTS

Reviewer #1 (Remarks to the Author):

The authors have provided strong evidence for characterizing isolated atomic structures and the reaction pathways, effectively addressing most of the concerns reviewers have.

I thank the authors for demonstrating scenarios where direct seawater electrolysis could be beneficial. It is logical to remove Ca^{2+} and Mg^{2+} from seawater to prevent the potential generation of insoluble compounds in the electrolyzer. However, it's important to note that the top five ions in seawater are Cl^- , Na^+ , Mg^{2+} , SO_4^{2-} , and Ca^{2+} (https://supremecourt.flcourts.gov/content/download/242684/file/16-576_Seawater.pdf). As time goes on, SO_4^{2-} concentration will keep increasing in the electrolyzer. Thus, I have one final question: considering that SO_4^{2-} could be absorbed on the catalyst surface, how might this affect the electrolyzer's performance? For example, SO_4^{2-} could repulse Cl^- ions from the electrode surface and prevent Cl^- adsorption (Angew. Chem. Int. Ed. 133, 22922-22926. (2021)).

Reviewer #2 (Remarks to the Author):

This article introduces an atomic Ir catalyst on cobalt-iron layered double hydroxide (Ir/CoFe-LDH) for seawater oxidation. In contrast to conventional catalysts designed to completely repel Cl^- adsorption for seawater electrolysis, the atomic Ir sites on CoFe-LDH allow Cl^- adsorption to modulate the electronic structure of the Ir active center, thereby establishing a unique Ir-OH/ Cl^- coordination in alkaline seawater electrolyte. However, the authors did not address the role of Co, Fe. Since, in the electrolyte, all Fe and Co participate in the reaction, questions arise about the stability over an extended period. Even though the authors responded to the reviewers' comments, this paper requires more revisions before being considered for publication in Nature Communications. Additionally, it needs clearer and improved research for better understanding.

1. Metal double layer hydroxides (MDLH), including layered double hydroxides (LDHs), are indeed known to have some limitations in terms of stability in seawater. These materials can undergo various processes, including dissolution, re-precipitation, and structural changes, when exposed to the complex and dynamic environment of seawater. The specific challenges or stability issues may vary depending on the composition of the MDLH, the presence of certain ions, and other environmental factors. Corrosion and Oxidation: Certain MDLHs may be susceptible to corrosion or oxidation in the presence of seawater, leading to degradation over time. So how can only Ir doped to CoFe-LDH improve stability.

2. 'Dynamic Chloride Ion Adsorption on Single Iridium Atom Boosts Seawater Oxidation Catalysis'; however, in the research, the authors fabricated Ir atom doped FeCo LDH, which is not clear. What is the

role of Co, Fe, and O in hybrid. the author explained in detail. Why the author did use other substrates such as Ni, Al...

3. The atomic Ir sites on CoFe-LDH allow for Cl⁻ adsorption to modulate the electronic structure of the Ir active center, thereby establishing a unique Ir-OH/Cl coordination in alkaline seawater electrolyte. In Ir-doped CoFe-LDH hybrids, the author must perform Density Functional Theory (DFT) calculations on the materials to identify active sites of Fe, Co, O, and Ir. However, in this study, the focus is solely on the interaction of Cl⁻ with Ir-OH, and other cases such as Cl-OH-Fe and Co-OH-Cl are not addressed.

4. To demonstrate the change in catalytic activity before and after the introduction of a single iridium atom into CoFe-LDH, the highest catalyst activity for the Oxygen Evolution Reaction (OER) is observed. The Density of States (DOS) and Total Density of States (TDOS) of CoFe-LDH and Ir-doped CoFe-LDH are analyzed. This analysis can explain the electrical conductivity, and the Projected Density of States (PDOS) is used to elucidate chloride ion absorption both with and without the presence of the iridium atom.

5. The authors must provide high-magnification SEM images of the nanosheets of CoFe LDH and Ir/CoFe LDH; and The SEM images of the materials after 1000 hours of stability testing should also depict their morphology.

6. To apply these materials in real industrial applications, Ir/CoFe LDH must undergo OER testing and stability assessment in seawater, with results presented in terms of overpotential. Additionally, characterization techniques such as XRD, SEM, TEM, and XPS should be employed to provide a comprehensive analysis.

7. To confirm the stability of the Ir atom on CoFe LDH, it is essential to verify not only its existence as a single atom but also in cluster formations. The authors should conduct thorough confirmation through large-area SEM and TEM analyses, supplementing the findings with additional image data.

8. In the electrolyte solution, Cl⁻ ions react with Fe and Co on the surface, resulting in instability and corrosion in a Cl⁻-ion-rich environment. Consequently, Ir may separate in the hybrid. The authors should conduct experiments to provide an explanation in such cases.

Response Letter

Reviewer #1 (Remarks to the Author):

The authors have provided strong evidence for characterizing isolated atomic structures and the reaction pathways, effectively addressing most of the concerns reviewers have.

I thank the authors for demonstrating scenarios where direct seawater electrolysis could be beneficial. It is logical to remove Ca^{2+} and Mg^{2+} from seawater to prevent the potential generation of insoluble compounds in the electrolyzer. However, it's important to note that the top five ions in seawater are Cl^- , Na^+ , Mg^{2+} , SO_4^{2-} , and Ca^{2+}

(https://supremecourt.flcourts.gov/content/download/242684/file/16-576_Seawater.pdf). As time goes on, SO_4^{2-} concentration will keep increasing in the electrolyzer. Thus, I have one final question: considering that SO_4^{2-} could be absorbed on the catalyst surface, how might this affect the electrolyzer's performance? For example, SO_4^{2-} could repulse Cl^- ions from the electrode surface and prevent Cl^- adsorption (Angew. Chem. Int. Ed. 133, 22922-22926. (2021)).

Reply: We appreciate the reviewer for the constructive comments. Considering the strong coordinating effect of Ir-Cl, the initial Ir surface is likely to adopt a stable Ir-OH/Cl coordination state, facilitating the occurrence of oxygen evolution reaction.

To investigate whether SO_4^{2-} would impact the Ir-Cl coordination, we conducted cyclic voltammetry (CV) tests on Ir/CoFe-LDH in a solution of 6 M NaOH + 2.8 M NaCl. Subsequently, we gradually introduced 0.03 M Na_2SO_4 (the sulfate ion concentration in seawater, as cited from ACS Energy Letters 4, 933-942 (2019)) and continued the CV tests. The results indicate virtually sustained performance (6 M NaOH + 2.8 M NaCl: 205 mV, 6 M NaOH + 2.8 M NaCl + 0.03 M Na_2SO_4 : 208 mV, a difference of 3 mV), affirming that SO_4^{2-} does not affect the catalyst's activity (as shown in Figure R1 and new Extended Data Fig. 28).

Figure R1. CV curves of Ir/CoFe-LDH recorded in 6 M NaOH + 2.8 M NaCl with different SO_4^{2-} concentrations (from 0.03 M, representing the sulfate concentration in seawater, to a tenfold enrichment of sulfate concentration, reaching 0.3 M).

To investigate whether SO_4^{2-} under cyclic conditions would displace coordinated Cl^- , we conducted a two-hour CV test at a scan rate of 50 mV/s in the potential range from 1.024 to 2.124 versus RHE. The results showed no discernible change in performance before and after the CV cycling (**Figure R2** and new **Extended Data Fig. 29**).

Figure R2. The CV curves of Ir/CoFe-LDH before and after a 2-hour cyclic voltammetry test in 6 M NaOH + 2.8 M NaCl + 0.03 M SO_4^{2-} .

Subsequently, more SO_4^{2-} was introduced into the electrolyte, eventually reaching a concentration of 0.3 M SO_4^{2-} (tenfold enrichment of sulfate concentration in real seawater). CV curves indicate that the catalyst's performance fluctuation is negligible (≤ 5 mV) within the SO_4^{2-} concentration range of 0.03 M ~ 0.3 M (**Figure R1** and new **Extended Data Fig. 28**). This suggests that after the formation of a stable Ir-Cl coordination state, the presence of SO_4^{2-} does not exert any discernible influence on the performance (Ir-Cl coordinates first due to the lower adsorption energy of Ir-Cl coordination, and the concentration of SO_4^{2-} is inconsequential).

The following discussion has been added to the “electrochemical performance of Ir/CoFe-LDH” section in the revised manuscript:

Page 12, line 264:

“To investigate whether SO_4^{2-} would impact the Ir-Cl coordination, we conducted cyclic voltammetry tests on Ir/CoFe-LDH in a solution of 6 M NaOH + 2.8 M NaCl. Subsequently, we gradually introduced 0.03 M Na_2SO_4 (the sulfate ion concentration in seawater¹) and continued the CV tests. The results indicate virtually sustained performance (within 3 mV), affirming that SO_4^{2-} does not affect the catalyst’s activity (Extended Data Fig. 28). To demonstrate whether SO_4^{2-} under cyclic conditions would displace coordinated Cl^- , we conducted a two-hour CV test, the results showed no discernible change in performance before and after the CV cycling (Extended Data Fig. 29). Subsequently, more SO_4^{2-} was introduced into the electrolyte, eventually reaching a concentration of 0.3 M SO_4^{2-} (tenfold enrichment of sulfate concentration in real seawater). CV curves indicate that the catalyst’s performance fluctuation is negligible (≤ 5 mV) within the range SO_4^{2-} concentration of 0.03 M ~ 0.3 M (Extended Data Fig. 28). This suggests that after the formation of a stable Ir-Cl coordination state, the presence of SO_4^{2-} does not exert any discernible influence on the performance.”

Reviewer #2 (Remarks to the Author):

This article introduces an atomic Ir catalyst on cobalt-iron layered double hydroxide (Ir/CoFe-LDH) for seawater oxidation. In contrast to conventional catalysts designed to completely repel Cl^- adsorption for seawater electrolysis, the atomic Ir sites on CoFe-LDH allow Cl^- adsorption to modulate the electronic structure of the Ir active center, thereby establishing a unique Ir-OH/ Cl^- coordination in alkaline seawater electrolyte. However, the authors did not address the role of Co, Fe. Since, in the electrolyte, all Fe and Co participate in the reaction, questions arise about the stability over an extended period. Even though the authors responded to the reviewers’ comments, this paper requires more revisions before being considered for publication in Nature Communications. Additionally, it needs clearer and improved research for better understanding.

1. Metal double layer hydroxides (MDLH), including layered double hydroxides (LDHs), are indeed known to have some limitations in terms of stability in seawater. These materials can undergo various processes, including dissolution, re-precipitation, and structural changes, when exposed to the complex and dynamic environment of seawater. The specific challenges or stability issues may vary depending on the composition of the MDLH, the presence of certain ions, and other environmental factors. Corrosion and Oxidation: Certain MDLHs may be susceptible to corrosion or oxidation in the presence of seawater, leading to degradation over time. So how can only Ir doped to CoFe-LDH improve stability.

Reply: We thank the reviewer for the valuable comments. The deactivation of LDH can be attributed to local pH reduction, interlayer acidification, cation dissolution, or oxidation of metal centers to high-valent cations leading to their leaching (Chem. Soc. Rev, 2021, 50, 8790-8817), and in seawater environment, the presence of chloride ions may exacerbate these risks.

In our investigation, after Ir loading onto CoFe LDH, we conducted DFT calculations for the Fe-

Room 40E, Zong-He Building, Beijing University of Chemical Technology, Beijing 100029, P. R. China

Cl and Co-Cl configurations, selecting Cl adsorption sites on Fe or Co, respectively. The results indicate that in the presence of Ir, Cl cannot effectively adsorb onto Fe or Co due to the competition of Ir (*see the GIF in Figure R3, attachment files, and new Extended Data Fig. 54*). The optimization process revealed direct Cl adsorption onto Ir sites, eliminating the existence of stable configurations like Fe-Cl or Co-Cl (**Figure R4 and new Extended Data Fig. 55**). Consequently, these two coordination forms are not competitive or stable. This finding further confirms the significant improvement in the stability of CoFe LDH upon the introduction of Ir. The strong Ir-Cl coordination stabilizes the CoFe sites, inhibiting Cl coordination with both metal sites, thus ensuring robust stability in seawater conditions.

Furthermore, XPS data from real seawater stability tests reveal that post-reaction (**Figure R5 and new Extended Data Fig. 37**), the oxidation state of Co and Fe show no significant change. This suggests that Ir-Cl coordination stabilizes the underlying CoFe sites, preventing an increase in oxidation state and leaching, thereby ensuring the stability of CoFe substrate.

Figure R3. The Cl movement trajectory GIF during the formation of stable configurations: from (left) Co-Cl to Ir-Cl, and from (right) Fe-Cl to Ir-Cl. (yellow: Fe, blue: Co, grey: Ir, green: Cl, red: O, pink: H).

Figure R4. Structure evolution during structural optimization/relaxation: from (a) Co-Cl to (b) Ir-Cl, and from (c) Fe-Cl to (d) Ir-Cl (yellow: Fe, blue: Co, grey: Ir, green: Cl, red: O, pink: H).

Figure R5. The XPS spectra of (a) Co, and (b) Fe after long-term stability test in real seawater.

The following discussion has been added to the “electrochemical performance of Ir/CoFe-LDH” and “Theoretical investigation” section in the revised manuscript:

Page 12, line 280

“The deactivation of LDH can be attributed to local pH reduction, interlayer acidification, cation dissolution, or oxidation of metal centers to high-valent cations leading to their leaching² and in seawater environment, the presence of chloride ions may exacerbate these risks.”

Page 20, line 453

“To further demonstrate the stability of Ir/CoFe-LDH, we selected Cl adsorption sites on Fe or Co, respectively. The results indicate that, in the presence of Ir, Cl cannot effectively adsorb onto Fe or Co, respectively (Extended Data Fig. 54). The optimization process revealed direct Cl adsorption onto Ir sites, eliminating the existence of stable configurations like Co-Cl or Fe-Cl (Extended Data Fig. 55&56). Consequently, these two active sites do not exist at presence of Ir. This finding further confirms the significant improvement in the electrochemical stability of CoFe LDH upon the introduction of Ir. The strong Ir-Cl coordination stabilizes the CoFe sites, inhibiting Cl coordination with both metal sites, thus ensuring system stability in seawater electrolysis.”

Page 13, line 309

“The XPS measurement of Ir/CoFe-LDH after long-term OER stability test (Extended Data Fig. 37) revealed that the oxidation state of Co and Fe showed no significant change, confirming that Ir-Cl coordination could stabilize CoFe-LDH, preventing oxidation and dissolution of CoFe-LDH, thereby ensuring the stability of CoFe-LDH substrate.”

2. Dynamic Chloride Ion Adsorption on Single Iridium Atom Boosts Seawater Oxidation Catalysis'; however, in the research, the authors fabricated Ir atom doped FeCo LDH, which is not clear. What is the role of Co, Fe, and O in hybrid. the author explained in detail. Why the author did use other substrates such as Ni, Al...

Reply: We appreciate the reviewer for the constructive comments. In our previous work (Nature Communications, <https://doi.org/10.1038/s41467-019-09666-0>), we successfully dispersed Ru single atoms by loading them onto CoFe-LDH, where the bivalent Co and trivalent Fe effectively dispersed the Ru single atoms. Following a similar approach, we loaded Ir as a single-atom catalyst, leveraging the excellent redox activity of CoFe and their strong synergistic effects with single atoms,

Room 40E, Zong-He Building, Beijing University of Chemical Technology, Beijing 100029, P. R. China

making them conducive for the dispersion of Ir.

Previous studies have provided us with valuable clues, that LDH elemental combinations with lower electronegativity exhibit enhanced electron coupling with metal single atoms. Based on electronegativity values (Fe (1.83) < Co (1.88) < Ni (1.92)), it is anticipated that noble metal single atoms on CoFe LDHs may demonstrate superior OER performance. Therefore, we continued to utilize CoFe LDH as the single-atom substrate, yielding highly favorable results.

The following discussion has been added to the “Synthesis and structural characterization of Ir/CoFe-LDH” section in the revised manuscript:

Page 4, line 112

“LDH elemental combinations with lower electronegativity may exhibit enhanced electron coupling with metal single atoms³. Based on electronegativity values (Fe (1.83) < Co (1.88) < Ni (1.92)), it is anticipated that noble metal single atoms on CoFe LDHs may demonstrate superior OER performance.”

3. The atomic Ir sites on CoFe-LDH allow for Cl⁻ adsorption to modulate the electronic structure of the Ir active center; thereby establishing a unique Ir-OH/Cl coordination in alkaline seawater electrolyte. In Ir-doped CoFe-LDH hybrids, the author must perform Density Functional Theory (DFT) calculations on the materials to identify active sites of Fe, Co, O, and Ir. However, in this study, the focus is solely on the interaction of Cl⁻ with Ir-OH, and other cases such as Cl-OH-Fe and Co-OH-Cl are not addressed.

Reply: We appreciate the reviewer for the constructive comments. We conducted additional DFT calculations for the FeOHCl and CoOHCl configurations, selecting Cl adsorption sites on Fe or Co, respectively. The results indicate that, in the presence of Ir, Cl cannot effectively adsorb onto Fe or Co, respectively (*see the GIF in Figure R6, attachment files, and new Extended Data Fig. 54*). The optimization process revealed direct Cl adsorption onto Ir sites, eliminating the existence of stable configurations like Co-Cl or Fe-Cl (**Figure R7&8 and new Extended Data Fig. 55&56**). Consequently, these two active sites do not exist at presence of Ir. This finding further confirms the significant improvement in the electrochemical stability of CoFe LDH upon the introduction of Ir. The strong Ir-Cl coordination stabilizes the CoFe sites, inhibiting Cl coordination with both metal sites, thus ensuring system stability in seawater electrolysis.

Room 40E, Zong-He Building, Beijing University of Chemical Technology, Beijing 100029, P. R. China

Figure R6. *The Cl movement trajectory GIF during the formation of stable configurations:* from (left) Co-Cl to Ir-Cl, and from (right) Fe-Cl to Ir-Cl. (yellow: Fe, blue: Co, grey: Ir, green: Cl, red: O, pink: H).

Figure R7. Structure evolution during structural optimization/relaxation: from (a) Co-Cl to (b) Ir-Cl, and from (c) Fe-Cl to (d) Ir-Cl (yellow: Fe, blue: Co, grey: Ir, green: Cl, red: O, pink: H).

Figure R8. Structure evolution during structural optimization/relaxation: from (a) Co-Cl to (b) Ir-Cl, and from (c) Fe-Cl to (d) Ir-Cl (yellow: Fe, blue: Co, grey: Ir, green: Cl, red: O, pink: H).

The following discussion has been added to the “Theoretical investigation” section in the revised manuscript:

Page 20, line 453

“To further demonstrate the stability of Ir/CoFe-LDH, we selected Cl adsorption sites on Fe or Co, respectively. The results indicate that, in the presence of Ir, Cl cannot effectively adsorb onto Fe or Co, respectively (Extended Data Fig. 54). The optimization process revealed direct Cl adsorption onto Ir sites, eliminating the existence of stable configurations like Co-Cl or Fe-Cl (Extended Data Fig. 55&56). Consequently, these two active sites do not exist at presence of Ir. This finding further confirms the significant improvement in the electrochemical stability of CoFe LDH upon the introduction of Ir. The strong Ir-Cl coordination stabilizes the CoFe sites, inhibiting Cl coordination with both metal sites, thus ensuring system stability in seawater electrolysis.”

Room 40E, Zong-He Building, Beijing University of Chemical Technology, Beijing 100029, P. R. China

4. To demonstrate the change in catalytic activity before and after the introduction of a single iridium atom into CoFe-LDH, the highest catalyst activity for the Oxygen Evolution Reaction (OER) is observed. The Density of States (DOS) and Total Density of States (TDOS) of CoFe-LDH and Ir-doped CoFe-LDH are analyzed. This analysis can explain the electrical conductivity, and the Projected Density of States (PDOS) is used to elucidate chloride ion absorption both with and without the presence of the iridium atom.

Reply: We appreciate the reviewer for the constructive comments. We conducted TDOS and PDOS calculations for two conditions: Ir-Cl/CoFe-LDH and Cl/CoFe-LDH. The results indicate that Cl hybridizes with the conduction bands of Co and Fe, but there is no hybridization with the valence bands (**Figure R9 b and new Extended Data Fig. 49 b**). Upon loading a Ir single atom, Ir-Cl exhibits significantly stronger hybridization with both the valence and conduction bands (**Figure R9 a and new Extended Data Fig. 49 a**), highlighting the enhanced orbital hybridization and bonding capability of Ir-Cl. The total density of states (TDOS) data aligns with the partial density of states (PDOS) data, demonstrating a pronounced increase in valence band hybridization in the presence of surface Ir, in contrast to CoFe-Cl case where the valence bands show no interaction (**Figure R10 and new Extended Data Fig. 50**).

Figure R9. PDOS of (a) Cl with Ir/CoFe-LDH and (b) Cl with CoFe-LDH.

Figure R10. TDOS of (a) Cl with Ir/CoFe-LDH and (b) Cl with CoFe-LDH.

Room 40E, Zong-He Building, Beijing University of Chemical Technology, Beijing 100029, P. R. China

The following discussion has been added to the “Theoretical investigation” section in the revised manuscript:

Page 19, line 432

“To further verify the specific role of Ir-Cl, we conducted calculations of the Density of States (DOS) and Projected Density of States (PDOS) for Ir-Cl/CoFe-LDH and Cl/CoFe-LDH. The results indicate that Cl hybridizes with the conduction bands of Co and Fe, but there is no hybridization with the valence bands (Extended Data Fig. 49b). Upon loading a Ir single atom, Ir-Cl exhibits significantly stronger hybridization with both the valence and conduction bands (Extended Data Fig. 49a), highlighting the enhanced orbital hybridization and bonding capability of Ir-Cl. The total density of states (TDOS) data aligns with the partial density of states (PDOS) data, demonstrating a pronounced increase in valence band hybridization in the presence of surface Ir, in contrast to CoFe-Cl case where the valence bands show no interaction (Extended Data Fig. 50)”

5. The authors must provide high-magnification SEM images of the nanosheets of CoFe LDH and Ir/CoFe LDH; and The SEM images of the materials after 1000 hours of stability testing should also depict their morphology.

Reply: We appreciate the reviewer for the constructive comments. We provided high-magnification SEM images of the nanosheets of CoFe-LDH and Ir/CoFe-LDH, as well as SEM images of the materials after 1000 hours of stability testing. The SEM images reveal that both CoFe-LDH and Ir/CoFe-LDH exhibit uniform nanosheet structures, with no noticeable changes in surface morphology upon Ir loading (**Figure R11 and new Extended Data Fig. 2**). After 1000 hours of stability test, Ir/CoFe-LDH maintains its roughly sheet-like structure (**Figure R12**).

Figure R11. High-magnification SEM images of (a) CoFe-LDH and (b) Ir/CoFe-LDH before OER test.

Room 40E, Zong-He Building, Beijing University of Chemical Technology, Beijing 100029, P. R. China

Figure R12. SEM image of Ir/CoFe-LDH after 1000 hours of stability test.

The following discussion has been added to the “Synthesis and structural characterization of Ir/CoFe-LDH” and “Electrochemical performance of Ir/CoFe-LDH” section in the revised manuscript:

Page 5, line 122

“SEM images reveal that both CoFe-LDH and Ir/CoFe-LDH exhibit uniform nanosheet structures, with no noticeable changes in surface morphology upon Ir loading (Extended Data Fig. 2).”

Page 13, line 300

“SEM after long-term stability test image revealed no significant changes in the morphology of the nanosheets (Extended Data Fig. 33)”

6. To apply these materials in real industrial applications must undergo OER testing and stability assessment in seawater, with results presented in terms of overpotential. Additionally, characterization techniques such as XRD, SEM, TEM, and XPS should be employed to provide a comprehensive analysis.

Reply: We appreciate the reviewer for the constructive comments. To realize seawater electrolysis in real industrial applications is a long journey to explore. We tried to demonstrate the feasibility of Ir/CoFe-LDH for such goal following the consequent ways:

- 1) We conducted CV tests in alkaline real seawater, revealing that Ir/CoFe-LDH exhibited an overpotential of only 208 mV at 10 mA cm⁻² (**Figure R13 and new Extended Data Fig. 30**), indicating its excellent performance under real industrial conditions. Subsequent CV tests conducted after stability tests at 24 hours, 200 hours, and 400 hours demonstrated that, following a 400-hour stability test, Ir/CoFe-LDH exhibited an overpotential of only 207 mV at 10 mA cm⁻² (**Figure R14 and new Extended Data Fig.31**). The current density increment was even higher than the initial performance, suggesting the stability of Ir/CoFe-LDH under realistic operating conditions.
- 2) XRD data before and after the stability test showed that Ir/CoFe-LDH maintained the characteristic peak of LDH at approximately 11.6° (**Figure R15 and new Extended Data Fig. 36**).
- 3) XPS data after long-term stability test in seawater indicated that the oxidation state of Co and Fe showed no significant change, suggesting that Ir-Cl could stabilize CoFe, inhibit CoFe oxidation and prevent CoFe leaching, ensuring the substrate’s stability (**Figure R16 and new Extended Data Fig. 37**).
- 4) TEM analysis after long-term stability test showed absence of particles or clusters (**Figure R17 and new Extended Data Fig. 34**), and SEM after long-term stability test revealed the persistence of a nanosheet structure in the substrate (**Figure R18 and new Extended Data Fig. 33**).
- 5) Moreover, it is noteworthy that we have already provided data for a 2000-hour stability test of Ir/CoFe-LDH in real seawater in response to the previous round of peer review comments (**Figure R19**).

All of the above characterizations demonstrated the good stability of Ir/CoFe-LDH in real seawater

Room 40E, Zong-He Building, Beijing University of Chemical Technology, Beijing 100029, P. R. China

electrolysis.

Figure R13. The CV curve of Ir/CoFe-LDH in 6 M NaOH + real seawater (calcium and magnesium ions are removed through pre-treatment).

Figure R14. The CV curves of Ir/CoFe-LDH before and after long-term stability test in 6 M NaOH + real seawater (calcium and magnesium ions are removed through pre-treatment).

Figure R15. The XRD patterns of Ir/CoFe-LDH after long-term stability test in real seawater.

Figure R16. The XPS spectra of (a) Co, (b) Fe, and (c) Ir after long-term stability test in real seawater.

Figure R17. The TEM image of Ir/CoFe-LDH after long-term stability test in real seawater.

Figure R18. The SEM image of Ir/CoFe-LDH after long-term stability test in real seawater.

Figure R19. The stability test of the Ir/CoFe-LDH//NiCoFeP electrolyzer in 6 M NaOH + real seawater (calcium and magnesium ions are removed through pre-treatment).

The following discussion has been added or clarified to the “Introduction”, and “Electrochemical performance of Ir/CoFe-LDH” section in the revised manuscript:

Page 3, line 65

“Moreover, Ir/CoFe-LDH required an overpotential of only 208 mV to achieve a current density of 10 mA cm⁻² in 6 M NaOH + seawater, indicating its excellent performance under real industrial conditions.”

Page 4, line 102

“Meanwhile, it can also operate stably in real seawater, reaching 10 mA cm⁻² with an

Room 40E, Zong-He Building, Beijing University of Chemical Technology, Beijing 100029, P. R. China

overpotential of 208 mV”

Page 12, line 288

“Ir/CoFe-LDH required an overpotential of only 208 mV to achieve a current density of 10 mA cm⁻² in a 6 M NaOH + seawater electrolyte (Extended Data Fig. 30), indicating its excellent performance under real industrial conditions. Additionally, CV tests conducted after stability tests at 24 hours, 200 hours, and 400 hours confirm the stability of Ir/CoFe-LDH under realistic operating conditions (Extended Data Fig. 31).”

Page 13, line 300

“SEM after long-term stability test image revealed no significant changes in the morphology of the nanosheets (Extended Data Fig. 33), while TEM analysis after long-term stability test image showed absence of clusters or particles (Extended Data Fig. 34).”

Page 13, line 307

“The XRD pattern of the post-OER Ir/CoFe-LDH (Extended Data Fig. 36) displays the characteristic peak of LDH at about 11.6°. The XPS measurement of Ir/CoFe-LDH after long-term OER stability test (Extended Data Fig. 37) revealed that the oxidation state of Co and Fe showed no significant change, confirming that Ir-Cl coordination could stabilize CoFe-LDH, preventing oxidation and dissolution of CoFe-LDH, thereby ensuring the stability of CoFe-LDH substrate. Furthermore, XPS quantitative analysis indicated a negligible change of the surface Ir concentration before and after OER.”

7. To confirm the stability of the Ir atom on CoFe LDH, it is essential to verify not only its existence as a single atom but also in cluster formations. The authors should conduct thorough confirmation through large-area SEM and TEM analyses, supplementing the findings with additional image data.

Reply: We appreciate the reviewer for the constructive suggestions. Accordingly, we have supplemented additional SEM and TEM data. TEM images show no discernible particles or clusters (**Figure R20 and new Extended Data Fig.34**), while SEM images indicate that Ir/CoFe-LDH maintains its nanosheet structure after the reaction (**Figure R21 new Extended Data Fig.33**). It is noteworthy that in the Supplementary Information, we have presented the HAADF-STEM and XAS data after the stability test (**Extended Data Fig. 28**). These data confirm the absence of Ir-Ir and Ir-O-Ir interactions, indicating that Ir did not form clusters or particles after the reaction.

Figure R20. Large-area TEM data of Ir/CoFe-LDH after long-term stability test in real seawater.

Room 40E, Zong-He Building, Beijing University of Chemical Technology, Beijing 100029, P. R. China

Figure R21. Large-area SEM data of Ir/CoFe-LDH after long-term stability test in real seawater.

The following discussion has been added to the “Electrochemical performance of Ir/CoFe-LDH” section in the revised manuscript:

Page 13, line 300

“SEM after long-term stability test image revealed no significant changes in the morphology of the nanosheets (Extended Data Fig. 33), while TEM analysis after long-term stability test image showed absence of clusters or particles (Extended Data Fig. 34).”

The following discussion has been highlighted in the “Electrochemical performance of Ir/CoFe-LDH” section in the revised manuscript:

Page 13, line 303

“HADDF-STEM characterizations (Extended Data Fig. 35) show that the distribution of atomic Ir atoms on CoFe-LDH surface after long-term stability test has no obvious change. There is still no Ir-Ir or Ir–O–Ir signal in the EXAFS data of post-OER catalyst, also revealing isolated dispersion of Ir atoms after long-term OER stability tests.”

8. In the electrolyte solution, Cl⁻ ions react with Fe and Co on the surface, resulting in instability and corrosion in a Cl⁻-ion-rich environment. Consequently, Ir may separate in the hybrid. The authors should conduct experiments to provide an explanation in such cases.

Reply: We appreciate the reviewer for the constructive comments. Firstly, through DFT calculations, we demonstrate that after the introduction of Ir, Cl⁻ exclusively adsorbs onto Ir, showing no interaction with CoFe, and there are no Co-Cl or Fe-Cl adsorption configurations (*see the GIF in Figure R22, attachment files, and new Extended Data Fig. 54*).

Additionally, XPS data reveal that the oxidation state of Co and Fe show no significant change after long-term stability test in real seawater (**Figure R23 and new Extended Data Fig. 37**), confirming that Ir-Cl coordination stabilizes the substrate, preventing oxidation and dissolution of CoFe, thereby ensuring the stability of CoFe substrate. Additionally, stabilized CoFe is further anchored with Ir through the strong synergistic effect, suppressing Ir leaching.

The dissolved Ir in electrolyte after OER stability test was quantified by ICP (9.558 ppb), which is nearly nine times less than the physical mixture of IrO₂ and CoFe-LDH (82.308 ppb) (**Table R1**). This experimental evidence further substantiates the stability of Ir loading on CoFe-LDH. Ir stabilizes CoFe through controllable Ir-Cl coordination, suppressing CoFe oxidation, in the meanwhile CoFe also stabilizes Ir through strong electronic interaction.

Room 40E, Zong-He Building, Beijing University of Chemical Technology, Beijing 100029, P. R. China

All these characterizations, coupled with the electrochemical data, collectively indicate the robust stability of Ir/CoFe-LDH in real seawater electrolysis.

Figure R22. The Cl movement trajectory GIF during the formation of stable configurations: from (left) Co-Cl to Ir-Cl, and from (right) Fe-Cl to Ir-Cl. (yellow: Fe, blue: Co, grey: Ir, green: Cl, red: O, pink: H).

Figure R23. High-resolution XPS spectra of (a) Co, and (b) Fe after long-term stability test in real seawater.

Table R1 Comparison of dissolved Ir in electrolyte after long-term stability test of Ir/CoFe-LDH and physical mixture of IrO₂ and CoFe-LDH with the same content of Ir.

Catalyst	Ir (ppb)
Ir/CoFe-LDH	9.558
IrO ₂ /CoFe-LDH	82.308

The following discussion has been added to the “Electrochemical performance of Ir/CoFe-LDH” and “Theoretical investigation” section in the revised manuscript:

Page 20, line 453

“To further demonstrate the stability of Ir/CoFe-LDH, we selected Cl adsorption sites on Fe or Co, respectively. The results indicate that, in the presence of Ir, Cl cannot effectively adsorb onto Fe or Co, respectively (Extended Data Fig. 54). The optimization process revealed direct Cl adsorption onto Ir sites, eliminating the existence of stable configurations like Co-Cl or Fe-Cl (Extended Data

Room 40E, Zong-He Building, Beijing University of Chemical Technology, Beijing 100029, P. R. China

Fig. 55&56). Consequently, these two active sites do not exist at presence of Ir. This finding further confirms the significant improvement in the electrochemical stability of CoFe LDH upon the introduction of Ir. The strong Ir-Cl coordination stabilizes the CoFe sites, inhibiting Cl coordination with both metal sites, thus ensuring system stability in seawater electrolysis.”

Page 13, line 309

“The XPS measurement of Ir/CoFe-LDH after long-term OER stability test (Extended Data Fig. 37) revealed that the oxidation state of Co and Fe showed no significant change, confirming that Ir-Cl coordination could stabilize CoFe-LDH, preventing oxidation and dissolution of CoFe-LDH, thereby ensuring the stability of CoFe-LDH substrate. Furthermore, XPS quantitative analysis indicated a negligible change of the surface Ir concentration before and after OER.”

The following discussion has been highlighted in the “Electrochemical performance of Ir/CoFe-LDH” section in the revised manuscript:

Page 13, line 314

“The dissolved Ir in the electrolyte after OER stability test was also quantified by inductively coupled plasma optical emission spectroscopy (ICP-MS, 9.558 ppb), which is nearly nine times less than the physical mixture of IrO₂ and CoFe-LDH (82.308 ppb), as shown in Extended Data Table 4. This experimental evidence further substantiates the stability of Ir loading on CoFe-LDH. Ir stabilizes CoFe through controllable Ir-Cl coordination, suppressing CoFe oxidation, in the meanwhile CoFe also stabilizes Ir through strong electronic interaction.”

References

1. Sören D. et al. Direct Electrolytic Splitting of Seawater: Opportunities and Challenges. *ACS Energy Lett.* 2019, 4, 4, 933–942.
2. Dao, J. et al. Layered double hydroxide-based electrocatalysts for the oxygen evolution reaction: identification and tailoring of active sites, and superaerophobic nanoarray electrode assembly. *Chem. Soc. Rev.*, 2021,50, 8790-8817.
3. Li, P. et al. Boosting oxygen evolution of single-atomic ruthenium through electronic coupling with cobalt-iron layered double hydroxides. *Nat Commun* 10, 1711 (2019).

REVIEWERS' COMMENTS

Reviewer #1 (Remarks to the Author):

I have carefully reviewed the manuscript and the accompanying rebuttal letter. The authors have convincingly demonstrated the stability of the catalysts using accelerated CV tests in electrolytes with concentrated SO_4^{2-} . Thus, I recommend the paper for publication in Nature Communications.

Reviewer #2 (Remarks to the Author):

The authors have addressed the reviewers' comments and revised the manuscript well. The revised manuscript is suitable for acceptance in Nature Communications.